

# One-class land-cover classification using MaxEnt: the effect of modelling parameterization on classification accuracy

Ignacio C. Fernández and Narkis S. Morales

Centro de Modelación y Monitoreo de Ecosistemas, Facultad de Ciencias, Universidad Mayor, Santiago, Chile

## ABSTRACT

Multiple-class land-cover classification approaches can be inefficient when the main goal is to classify only one or a few classes. Under this scenario one-class classification algorithms could be a more efficient alternative. Currently there are several algorithms that can fulfil this task, with MaxEnt being one of the most promising. However, there is scarce information regarding parametrization for performing land-cover classification using MaxEnt. In this study we aimed to understand how MaxEnt parameterization affects the classification accuracy of four different land-covers (i.e., built-up, irrigated grass, evergreen trees and deciduous trees) in the city of Santiago de Chile. We also evaluated if MaxEnt manual parameterization outperforms classification results obtained when using MaxEnt default parameters setting. To accomplish our objectives, we generated a set of 25,344 classification maps (i.e., 6,336 for each assessed land-cover), which are based on all the potential combination of 12 different classes of features restrictions, four regularization multipliers, four different sample sizes, three training/testing proportions, and 11 thresholds for generating the binary maps. Our results showed that with a good parameterization, MaxEnt can effectively classify different land covers with *kappa* values ranging from 0.68 for deciduous trees to 0.89 for irrigated grass. However, the accuracy of classification results is highly influenced by the type of land-cover being classified. Simpler models produced good classification outcomes for homogenous land-covers, but not for heterogeneous covers, where complex models provided better outcomes. In general, manual parameterization improves the accuracy of classification results, but this improvement will depend on the threshold used to generate the binary map. In fact, threshold selection showed to be the most relevant factor impacting the accuracy of the four land-cover classification. The number of sampling points for training the model also has a positive effect on classification results. However, this effect followed a logarithmic distribution, showing an improvement of *kappa* values when increasing the sampling from 40 to 60 points, but showing only a marginal effect if more than 60 sampling points are used. In light of these results, we suggest testing different parametrization and thresholds until satisfactory *kappa* or other accuracy metrics values are achieved. Our results highlight the huge potential that MaxEnt has a as a tool for one-class classification, but a good understanding of the software settings and model parameterization is needed to obtain reliable results.

Corresponding authors
Ignacio C. Fernández,
ignacio.fernandez@umayor.cl
Narkis S. Morales,
narkis.morales@umayor.cl

## INTRODUCTION

Land-cover analysis has become a fundamental approach for understanding, modelling and monitoring the spatio-temporal patterns of ecosystems worldwide (*Foley et al., 2005*; *Ellis et al., 2010*; *Pielke et al., 2011*). Land-cover analysis relies on a variety of land classification techniques, mostly based on computational algorithms, that commonly use remote sensing images as the main input for discriminating among different land-cover classes (*Rogan & Chen, 2003*; *Srivastava et al., 2012*). These classification techniques are aimed at translating the data contained in the images into usable information, such as land-cover maps. However, as in any modelling procedure, the quality and usefulness of the final outcome will depend on the particular technique used for discriminating among classes of interest (*Lu & Weng, 2007*).

One of the most common techniques for land-cover classification is based on the Maximum Likelihood (ML) algorithm, which is a supervised parametric pixel-based algorithm that is available in most of current GIS or GIS-capable platforms, such as QGIS, ArcGIS and R. The ML classification assumes that pixel values from a given layer will be normally distributed, therefore each pixel can be classified into a given class based on the maximum likelihood to pertain to the probability density function generated by the samples from that class (*Strahler, 1980*). Nevertheless, because ML is based on parametric data, it has shown to be outperformed by newer non-parametric algorithms that do not require the assumption of an *a-priori* particular distribution or number of parameters, such as Neural Network and Support Vector Machines (*Khatami, Mountrakis & Stehman, 2016*). Also, ML-based classifications often require to train the algorithm on every single class covering the study area, implying that many times resources and efforts will be placed on classifying classes that are not relevant for the study objectives (*Sanchez-Hernandez, Boyd & Foody, 2007*; *Li & Guo, 2010*; *Deng et al., 2018*). In fact, multi-class oriented classifiers, such as ML, may attempt to optimize the classification accuracy of all cover classes rather than focusing on increasing the accuracy of the target class (*Sanchez-Hernandez, Boyd & Foody, 2007*). Therefore, the use of one-class classification algorithms could provide a more efficient alternative than multi-classes approaches when the objective is classifying only one or a few classes of interest (*Deng et al., 2018*).

Among one-class classification algorithms, MaxEnt is probably one of the most used nowadays. MaxEnt is a maximum entropy algorithm-based software that was originally developed to be used for species distribution modelling (*Phillips & Dudík, 2008*; *Elith et al., 2011*). Nevertheless, its built-in algorithm has also been used for one-class classification of a variety of land-use or -cover types. For example, MaxEnt has been used for mapping urban land-use classes in California (*Li & Guo, 2010*), urban land in China (*Lin et al., 2014*), cover of invasive plant species in Colorado (*Evangelista et al., 2009*) and California (*Skowronek, Asner & Feilhauer, 2017*), and conservation habitats (*Stenzel et al., 2014*) and

raised bogs in Southern Germany (*Mack et al., 2016*). The increasing popularity of MaxEnt for one-class land-cover classification is probably due to the fact that is a freely available software (https://biodiversityinformatics.amnh.org/open_source/maxent), has shown as good or better performance than other one-class classification methods (e.g., *Li & Guo, 2010*; *Mack et al., 2016*), and comes with default parameters settings that facilitates its operation (*Mack & Waske, 2017*).

Nevertheless, while the availability of predefined default parameters may have promoted MaxEnt massification, the use of this feature can also result in suboptimal land-cover classification outcomes. For example, recent studies from species distribution modelling show that MaxEnt default parameters seldom generate optimal outcomes (*Shcheglovitova & Anderson, 2013*; *Syfert, Smith & Coomes, 2013*; *Morales, Fernández & Baca-González, 2017*). In relation of using MaxEnt default parameter for land-cover classification, the few published studies of which we are aware, also report that MaxEnt tend to produce better classification outcomes when users select the optimal parameters (*Mack & Waske, 2017*; *Skowronek et al., 2018*).

There are several default parameters that are modifiable in MaxEnt. The two main modifiable parameters are feature classes (FC) and regularization multiplier (RM). FC corresponds to a set of mathematical transformation of the different covariates used in the model, while RM refers to a numerical parameter that reduces or increases the smoothness of the model (*Elith et al., 2011*; *Merow, Smith & Silander, 2013*). Other additional settings, such as sample size (SS), test percentage (TP) and binary threshold (BT), could also influence the final classification outcome. SS refers to the number of samples used for building the model. Although MaxEnt seems to be only slightly sensitive to sample size, larger sample sizes has shown to produce better modeling results (*Wisz et al., 2008*). TP refers to the proportion (or percentage) of the sample size used for testing the model, which will be the inverse of the proportion used for training. Therefore, TP can affect the model results by indirectly modifying the model sample size. BT corresponds to the specific threshold value used for transforming the continuous probabilistic outputs values into a binary (positive/negative) classification results. The chosen threshold could largely affect the classification results, nevertheless finding an optimal BT for land-cover classification is still one of the main challenges for using MaxEnt as a one-class classification algorithm (*Mack et al., 2016*; *Mack & Waske, 2017*).

Even though MaxEnt has shown promising results as a tool for one-class classification, there is still a big knowledge gap regarding how classification results are affected by the parameter settings used for modeling. Thus, in this work we aim to: (1) assess the effects of five different parameter settings (i.e., FC, RM, SS, TP, BT) combinations on MaxEnt-based land-cover classification outcomes; (2) evaluate if these effects depend on the land-cover under analysis; (3) identify potential best parameter setting combination for different land-cover classes and compare its classification accuracy with the outcomes of a default parameter setting classification.

## MATERIAL AND METHODS

### Study area

We developed our study within the metropolitan area of Santiago, Chile (33.4489°S, 70.6693°W). Santiago is the capital and largest urban area of Chile, harboring an estimated population of 5.3 million inhabitants (*Instituto Nacional de Estadísticas, 2018*). The urban area covers a surface of 875 km$^2$ (*De la Barrera & Henríquez, 2017*), with an elevation ranging from 450 to 1,000 m above sea level. The climate within the study area is classified as Mediterranean, with marked colder temperatures and rainy periods during winter months, and warmer and dry period during summer (*Luebert & Pliscoff, 2006*). Historical annual precipitations average ∼340 mm, with more than 88% falling from May to September, and less than 2% precipitates during the three warmest months (December to February). Seasonal variations in temperatures and precipitation generate noticeable phenological changes in Santiago's urban vegetation, making it feasible to discriminate among different vegetation types from airborne or spaceborne images (*Fernandez & De la Barrera, 2018*).

### Remote sensing layers

To perform the land-cover classification we used Sentinel-2 mission satellite images with 0% cloud cover, representing vegetation conditions of summer (March 06, 2016) and winter (August 02, 2016) for the study area. To increase the resolution of our analysis we only used Sentinel-2 bands with a native resolution of 10 m/pixel (i.e., red, green, blue, NIR). Based on these bands, we also calculated the Normalized Difference Vegetation Index (NDVI) for summer and winter. We also generated an additional layer based on the arithmetic difference between the summer and winter NDVI (i.e., seasonal difference of NDVI). Even though MaxEnt is regarded to be only slightly sensitive to potential over-parameterization issues (*Warren & Seifert, 2011*), it is recommended to minimize correlation among predictor variables (*Merow, Smith & Silander, 2013*). Therefore, we performed a Pearson correlation analysis for the 11 layers (i.e., red, green, blue, and NIR, for summer and winter, and the three NDVI layers), and we kept only those layers with correlations smaller than 0.8. After this analysis we ended up with seven layers: Red (summer/winter), NIR (summer/winter), NDVI (summer/winter) and the seasonal difference of NDVI. All remote-sensing layers procedures were done in QGIS 2.18 Las Palmas (http://www.qgis.org).

### Land-cover classification (Maxent modelling)

We assessed Maxent one-class classification performance by focusing on four main land-cover types present in Santiago: (1) Built-up infrastructure, represented by roads and other paved structures, houses, buildings and commercial infrastructure; (2) irrigated grass, mostly associated to private and public lawns, sport fields and other summer irrigated grasses; (3) evergreen trees, related to native and exotic urban shrubs and trees with all year round leaves and photosynthetic activity; (4) deciduous trees, mostly winter deciduous exotic shrubs and trees planted on residential gardens, streets and urban parks.

To perform the classification modelling we started by selecting 100 ground-true points for each land-cover class by using very-high resolution images (<1 m/pixel) available

through Google Earth. Because the potential occurrence of spatial mismatches between Google and Sentinel images due to georeferencing and resolution differences, we only selected points that were within a 5 m radius of the type of land-cover under analysis (i.e., minimum size 1 pixel). Each set of points was doubled checked independently by each author. To assess the effect of MaxEnt parametrization on land-cover classification, we generated a set of 25,344 classification maps (i.e., 6,336 for each of the four assessed land-covers), which are based on all the potential combination of the five analyzed parameters (Table 1). These combinations included first, 12 different types of restrictions ("feature class") in the modelling stage based on the combination of lineal (L), hinge (H), quadratic (Q), threshold (T) and product (P). We used all the computational possible combinations of these features. Second, four regularization multipliers (i.e., 0.25, 1, 3 and 5). Third, four different sample sizes (i.e., 40, 60, 80 and 100). For the later, samples smaller than 100 were randomly sampled from the original 100 sampling points. Fourth, three different percentages of testing proportion to validate the model (i.e., 10, 30, 50). Fifth, 11 binary thresholds for generating the final binary classification maps: Fixed cumulative value 1 threshold (FC1); fixed cumulative value 5 threshold (FC5); fixed cumulative value 10 threshold (FC10); 10 percentile training presence threshold (10PTP); balance training omission, predicted area and threshold value threshold (BTOPA); equate entropy of thresholded and original distributions threshold (EETOD); equal test sensitivity and specificity threshold (ETeSS); equal training sensitivity and specificity threshold (ETrSS); maximum test sensitivity plus specificity threshold (MTeSPS); minimum training presence threshold (MTP); maximum training sensitivity plus specificity threshold (MTrSPS) (Table 1). All MaxEnt models were run using the random seed feature and replicated 10 times, and we used the average values of these replicates as the modeling outcomes. Background points were set to default (maximum 10,000 points). All modelling procedures were performed using MaxEnt 3.4.1 (https://biodiversityinformatics.amnh.org/open_source/maxent). Application of thresholds and generation of final binary maps were done in R version 3.5.1 (R Core Team, 2018).

## Classification accuracy

We evaluated the accuracy of the classification result by estimating the *kappa* coefficient. To calculate *kappa,* we randomly sampled 1,000 points within a quadrant of $16 \times 16$ km, which is the largest square that can be drawn within the convoluted shape of Santiago. We decided to use this quadrant to avoid potential undesirable effects of choosing sampling points outside Santiago's urbanized area and to reduce the size of files to increase the efficiency of the computational processes. Each of these 1,000 points were visually classified into built-up, irrigated grass, evergreen trees, deciduous trees and "other class" by using very-high resolution images through Google Earth. If a sampling point fell on mixed land-cover (e.g., transition between two or more land covers) we moved the point to the closest single-land cover surface representing the dominant land-cover from the original location. We considered mixed land-covers all points falling in locations with more than one land-cover within a 5 m radius. All points were double checked by the authors. We

**Table 1  Parameters used to generate the land-cover classification results.** See methodological section for references to the acronym used in FC and BT columns.

| Feature class (FC) | Regul. multiplier (RM) | Sample size (SS) | Test percentage (TP) | Binary threshold (BT) |
|---|---|---|---|---|
| L | 5.00 | 100 | 50 | FC1 |
| H | 3.00 | 80 | 30 | FC5 |
| Q | 1.00 | 60 | 10 | FC10 |
| T | 0.25 | 40 | | 10PTP |
| LQ | | | | BTOPA |
| PT | | | | EETOD |
| QH | | | | ETeSS |
| QP | | | | ETrSS |
| TQ | | | | MTeSPS |
| LQP | | | | MTP |
| LQPT | | | | MTrSPS |
| LQPTH | | | | |

then used the 1,000 sampling points to build four different testing layers (one for each assessed land-cover) based on positive and negative labels for each land-cover. We used these testing layers to calculate the *kappa* coefficient for each classification result based on the proportion of true positives, false positives, true negatives and false negatives, following the formula shown in *Fielding & Bell (1997)*.

We compared the classification accuracy of the different combinations through integrated violin-boxplot graphs for the *kappa* coefficient, which facilitates the visual comparison of differences on data distribution, interquartile and median values. All classification accuracy procedures and graphs were done in R version 3.5.1 (*R Core Team, 2018*).

## Comparison of default parameter vs best parameters settings

We defined the best combination of parameters for each land-cover type by looking at the *kappa's* median values of the violin-boxplots of feature classes, regularization multipliers, sample size and test percentage (See Figs. 1–4). Based on these results, we decided to use a SS of 100 sampling points and a TP of 10% for all classification models (i.e., default and best parameter settings), only modifying those parameters directly related to the modeling process, which are the FC and RM. Therefore, for the default parameter setting models we used the "autofeature" FC with a RM of 1, whereas for the best parameter setting we used the FC and RM showing the highest median *kappa* for each land-cover type (See Figs. 1 and 2). Using these parameters, we ran 30 replicates for each classification model in MaxEnt. To generate the final binary maps in R, we did not select a specific binary threshold because we did not know the effect of this parameter on the default setting models results. Instead, we compared modelling results by using all the BT provided in MaxEnt. We used a conventional student *t*-test to evaluate statistically significant differences between the default and best parameter settings for each land-cover type and binary threshold.
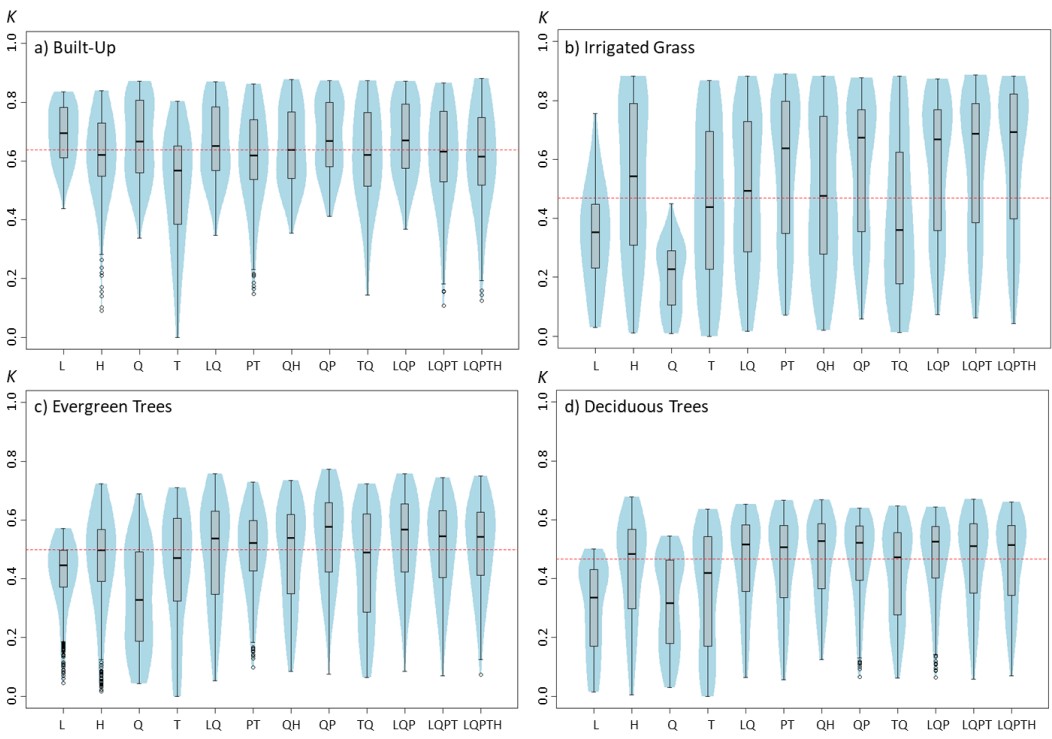

**Figure 1** **Violinplots of the *kappa* values obtained from the single land-cover maps generated through MaxEnt using the feature classes (FC) shown in the *x*-axis.** (A) Built-up; (B) irrigated grass; (C) evergreen trees; (D) deciduous trees. Each violinplot includes the combination of all the other tested parameters (See Table 1 for reference). The red dashed line is the median *kappa* value for all combinations.

## RESULTS

### Parameterization effects on classification accuracy

Results show that changing the feature classes can have important effects on the final land-cover classification accuracy, and that these effects change depending on the land-cover that is being classified (Fig. 1). For example, while for built-up land cover the L and Q features produced the highest *kappa* (measured as *kappa's* median value), these same features produced the poorest results for the three vegetated covers (i.e., irrigated grass, evergreen trees and deciduous trees). Furthermore, the FC for which the *kappa* median was largest, fluctuated for the four land-covers: L for built-up, LQPTH for irrigated grass, QP for evergreen trees and QH for deciduous trees, with median *kappa's* of 0.695, 0.693, 0.577 and 0.528, respectively. There are also important differences in the variability (i.e., density distribution) of *kappa* results associated to the different FC. For example, based on the differences between the first and third quartile, built-up, evergreen trees and deciduous trees covers show the largest dispersion of kappa values for T, Q, and TQ, but for irrigated grass all FC show very large dispersion, except for L and Q (See Fig. 1).

Modifying the regularization multiplier could also have important effects on classification results, but as our results show, these effects could considerably differ depending on the land-cover type being classified (Fig. 2). While for built-up, increasing

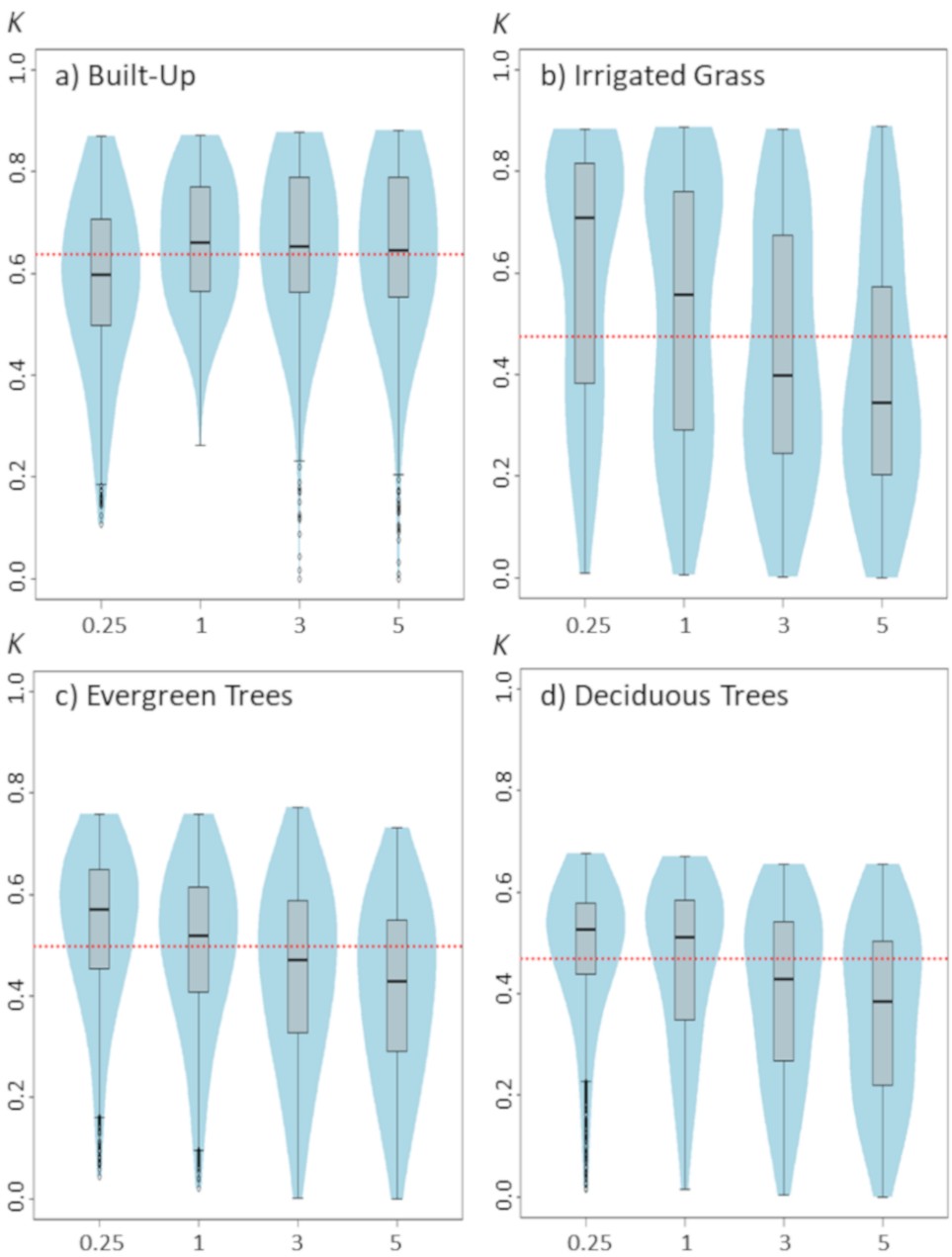

**Figure 2** **Violinplots of the *kappa* values obtained from the single land-cover maps generated through MaxEnt using the regularization multipliers (RM) shown in the *x*-axis.** (A) Built-up; (B) irrigated grass; (C) evergreen trees; (D) deciduous trees. Each violinplot includes the combination of all the other tested parameters (See Table 1 for reference). The red dashed line is the median *kappa* value for all combinations.

the RM from 0.25 to 1 seems to slightly improve classification results, the opposite pattern is observed for the three vegetated covers. In addition, while further increases (>1) in RM do not affect classification outcomes for built-up cover, they do affect the classification for vegetated covers, as is shown by the consistent decrease in *kappa* with larger RM. Among

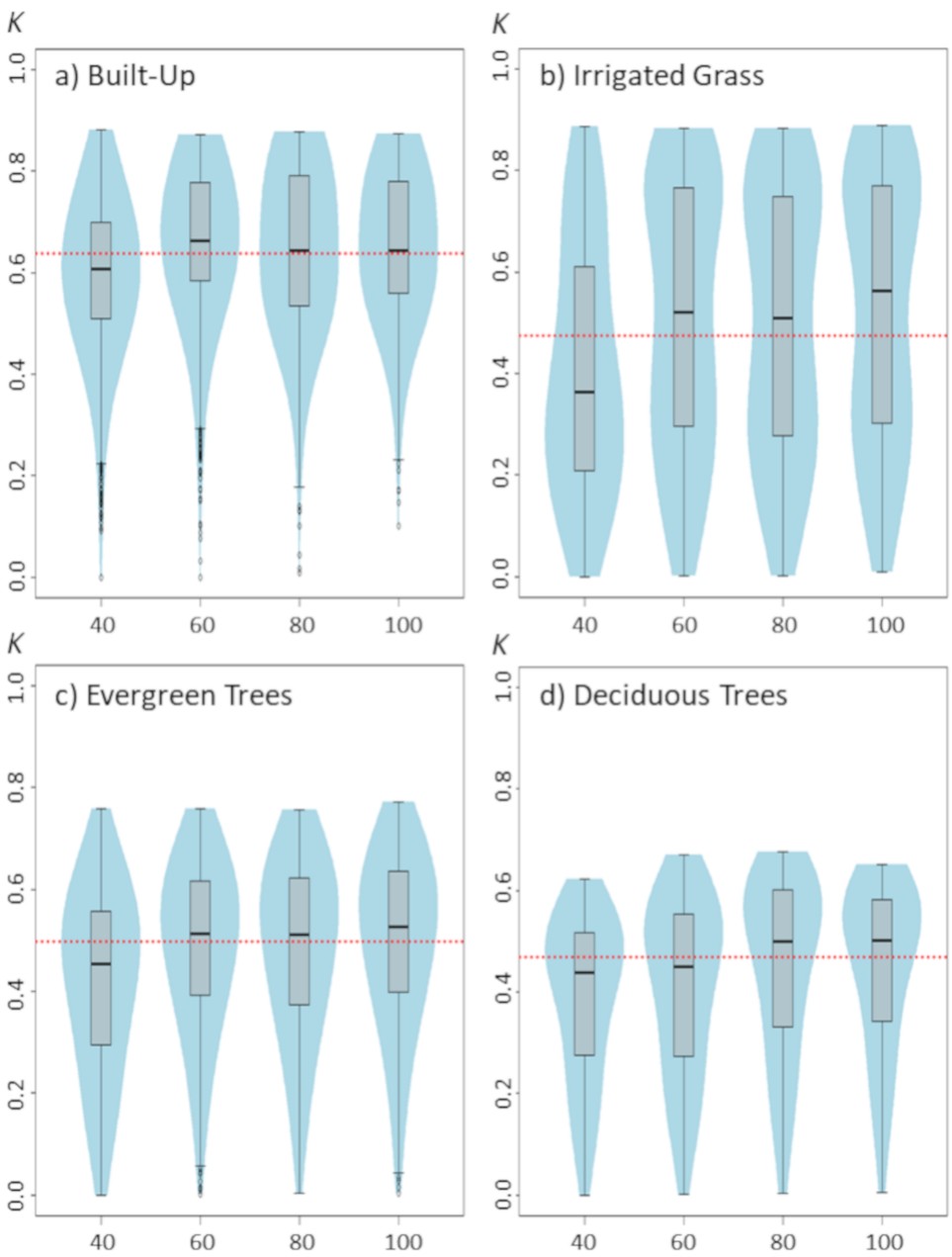

**Figure 3** **Violinplots of the *kappa* values obtained from the single land-cover maps generated through MaxEnt using the sample size (SS) shown in the *x*-axis.** (A) Built-up; (B) irrigated grass; (C) evergreen trees; (D) deciduous trees. Each violinplot includes the combination of all the other tested parameters (See Table 1 for reference). The red dashed line is the median *kappa* value for all combinations.

the three vegetated covers assessed in this work, irrigated grass cover showed by far the largest response to changes in RM (Fig. 2).

Changes in sample size show only a slight effect on classification results (Fig. 3). Although there are some differences between the four assessed land-covers, there is a consistent pattern showing that the lowest SS (i.e., 40 samples) generated the poorest

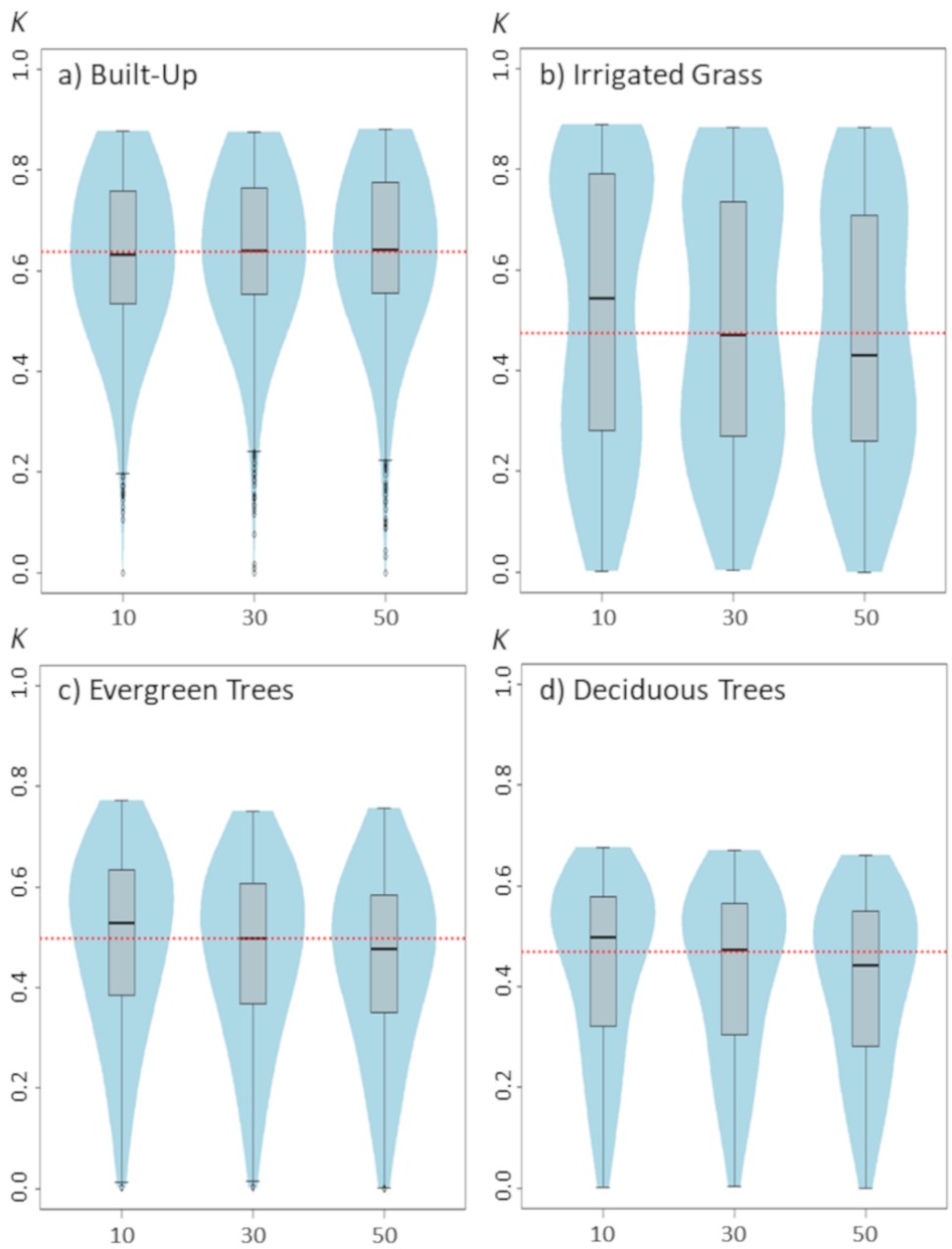

**Figure 4** **Violinplots of the *kappa* values obtained from the single land-cover maps generated through MaxEnt using the test percentage (TP) shown in the *x*-axis.** (A) Built-up; (B) irrigated grass; (C) evergreen trees; (D) deciduous trees. Each violinplot includes the combination of all the other tested parameters (See Table 1 for reference). The red dashed line is the median *kappa* value for all combinations.

classification results. For all assessed land covers, increasing the SS from 40 to 60 improved the classification. However, results show that increases in SS above 60 samples do not necessarily generate large improvements on classification results (Fig. 3).

Changing the test percentage for validating the model has only weak effects on classification results. In fact, results show that for built-up cover this setting seems to
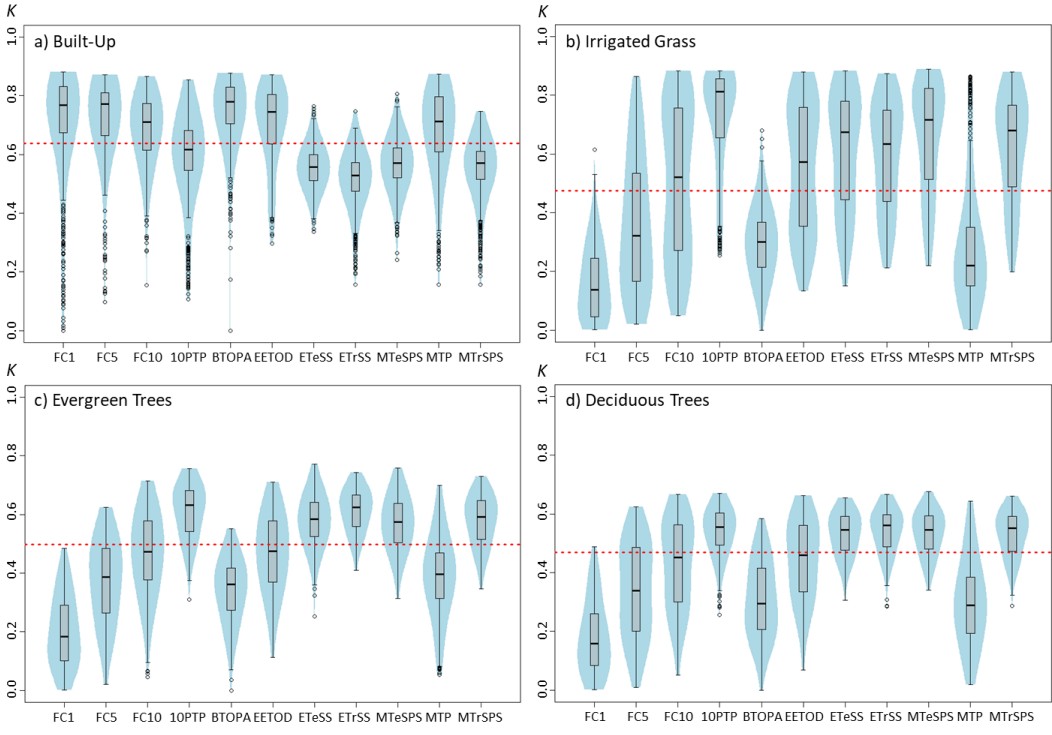

**Figure 5** **Violinplots of the *kappa* values obtained from the single land-cover maps generated through MaxEnt using the binary threshold (BT) shown in the *x*-axis.** (A) Built-up; (B) irrigated grass; (C) evergreen trees; (D) deciduous trees. Each violinplot includes the combination of all the other tested parameters (See Table 1 for reference). The red dashed line is the median *kappa* value for all combinations.

not affect the median value of *kappa* nor the *kappa's* median distribution, whereas for evergreen trees and deciduous trees it is observable a slight, but consistent, decrease in classification results as TP increase to 50% (Fig. 4). The largest effect of TP is shown for irrigated grass cover, which is consistent with the results shown for changes in the SS setting.

The selection of binary threshold values is the setting that had the largest effects on classification results among the five assessed settings (Fig. 5). Using one BT instead of another could greatly affect the classification results. For example, depending on the selected threshold, *kappa's* median for irrigated grass cover could range from 0.134 to 0.811 (Fig. 5B). The effect of BT on classification shows similar patterns for irrigated grass, evergreen trees and deciduous trees, but not for built-up cover. For instance, while for built-up cover the FC1 threshold generated one of the best classification results (*kappa's* median = 0.766), for the three vegetated covers this same threshold produced the worst results. At the other hand, 10PTP produced the best classification results for irrigated grass and evergreen trees, and the second best for deciduous trees, but produced poor classification results for built-up cover (Fig. 5).

**Default parameter vs. best parameters setting classification accuracy**

The comparison of the models produced using the default parameter setting for FC and RM with those using the identified best parameters (defined as FC and RM having the largest *kappa* median for each land-cover type, i.e., FC = L, RM = 1 for built-up; FC = LQPTH, RM = 0.25 for irrigated grass; FC = QP, RM = 0.25 for evergreen trees; FC = QH, RM = 0.25 for deciduous trees; See Figs. 1 and 2) show that the identified best parameter setting does not always outperform the classification accuracy of the default model (Fig. 6). Considering the four assessed land-cover types and the 11 thresholds, the best parameters outperformed the default parameters in 23 out of the 44 classification result comparisons (53,3%), while the identified best parameter was outperformed by the default parameter in 11 cases (25,0%). In 10 comparisons (22,7%) there were no statistical differences between both parameters settings (Fig. 6). The difference in classification results between the default and identified best parameters is highly dependent on the land-cover under analysis and the threshold used for generating the binary maps. For example, for evergreen trees, in nine out of 11 comparison the best parameter setting outperformed the default model and for no threshold the default model resulted in better classification. On the other hand, for irrigated grass the default model outperformed the identified best parameter setting in five out of 11 comparisons, while the opposite result was observed for four thresholds (Fig. 6). In general, BT selection has a much larger effect than parameter settings in classification results.

## DISCUSSION

Results from our work add to previous studies showing the potential of MaxEnt to be used as a one-class land-cover classification method (e.g., *Li & Guo, 2010*; *Amici, 2011*; *Lin et al., 2014*; *Stenzel et al., 2014*), but also draw attention to the importance of selecting best parameters combinations to achieve good classification accuracies (*Mack & Waske, 2017*). While our main objective was to understand how MaxEnt parameterization may affect classification results, we were also able to test the capability of MaxEnt for one-class land-cover classification, which has never been tested in an entire urban area the size of Santiago.

Our classification accuracy shows *kappa* values above 0.6 for all the assessed land-covers, and even above 0.8 for built-up and irrigated grass. These *kappa* values are comparable to those obtained by *Li & Guo (2010)* using Maxent to classify similar land-covers in a urban plot in a residential area of California, and also to those obtained by *Deng et al. (2018)* using other state of the art classification algorithms to classify land covers in a similar urban setting in California. Thus, results from our study highlight the potential of MaxEnt and its built-in maximum entropy algorithm to be used for one-class classification in complex spatially heterogeneous settings, such as urban areas.

**Parameterization effects on classification accuracy**

Results from our work show that model parameterization plays an important role on MaxEnt-based land-cover classification accuracy, and therefore an appropiate parameterization is highly relevant for achieving good classification results. However,

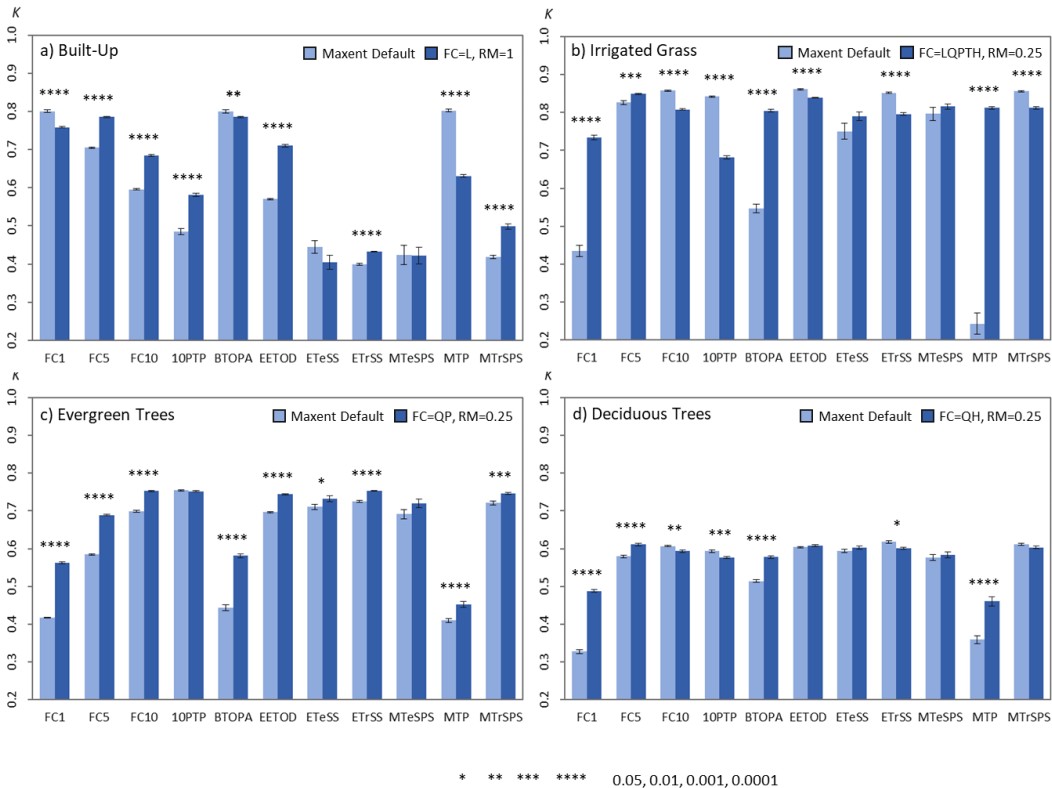

* ** *** ****   0.05, 0.01, 0.001, 0.0001

**Figure 6 Barplots comparing the average *kappa* values obtained from the single land-cover maps generated through MaxEnt using the default vs. best parameters settings.** (A) Built-up; (B) irrigated grass; (C) evergreen trees; (D) deciduous trees. Bars show the mean (±SE) *kappa* value of 30 replicates after applying the 11 BT. All models were built using a SS of 100 and a TP of 10%. The best FC and RM parameter setting used for each land-cover type is shown in the figure legend. Statistically significant values are represented by *p*-values of *(< 0.05), **(< 0.01), ***(< 0.001), ****(< 0.0001), after applying a two-sided *t*-test for each comparison (i.e., default vs. best parameter setting).

based in our results, is not clear if there are some specific feature classes that can provide higher quality classifications. On the contrary, it seems that the optimal feature classes to use will depend on the intrinsic spatial variability of the layers used to build the models. For example, in our study, the Q and L (i.e., quadratic and lineal) feature classes generate among the best classification accuracy for built-up, but the poorest results for vegetated covers. This result could be explained because the Q feature constrains the variation of the predictor variable to that of the sampling points of each layer (See *Elith et al., 2011*; *Merow, Smith & Silander, 2013* for an explanation of MaxEnt features). Thus, the Q feature may differentially affect the specificity of the resulting model depending on the true variance of the population, benefiting classification results of spatially homogenous covers, such as built-up, but weakening results for more heterogeneous covers, such as irrigated grass. A similar interpretation could be made for the L feature, as this feature simply uses the mean of the sampling points to estimate the conditions making a pixel suitable to be classified into a given class. These results suggest that MaxEnt models based on simpler parameter

combination could be well suited for classifying land-covers with more homogenous reflectance behaviors, such as built-up areas, while models of higher complexity would be needed for vegetated or other more heterogeneous areas. This could be further supported by the results of the regularization multiplier, that show that for vegetated land-covers, accuracy increases if regularization multiplier is reduced to fit more complex models, whereas built-up classification accuracy benefits from larger regularization that smooths out the model fit.

While sample size may not be directly considered a model parameter, it can have a direct effect on MaxEnt behavior, particularly if default features are used. In fact, MaxEnt will restrict the use of potential feature classes depending of the number of sampling points (*Elith et al., 2011*; *Merow, Smith & Silander, 2013*), therefore models generated with a small number of samples will be restricted in their complexity. This may explain why classification of the three vegetated land-covers are more affected than built-up cover by the number of sampling points used for training the model. Nevertheless, for the four land-covers, an effective sampling size of ∼80 seems to be optimal, which is consistent with the sampling size threshold at which MaxEnt is set to use all potential FC (*Elith et al., 2011*).

A key result from our study was that the selection of the binary threshold for building the classified maps is perhaps the most crucial factor related to the accuracy of the final classification outcome. This threshold is not a MaxEnt modeling parameter, but rather *a posteriori* decision that needs to be taken based on modeling results. While MaxEnt provides 11 potential thresholds (the 11 used in this study), the final decision on what threshold to use will mostly depend on the researcher or group of researchers performing the classification. Previous studies have pointed out to the difficulty and lack of information for finding an optimal threshold to maximize the classification accuracy using MaxEnt (*Li & Guo, 2010*; *Mack et al., 2016*; *Mack & Waske, 2017*), and based in our results it seems that this quest is still open. In fact, none of the tested thresholds performed well for all the assessed land-covers, implying that users will need to explore a set of thresholds to find one that produce optimal classification accuracy for each land-cover.

## Default parameter vs. best parameters setting classification accuracy

There is increasing evidence that using best, instead of default parameters, could improve the predictability of species distribution models produced through MaxEnt (*Shcheglovitova & Anderson, 2013*; *Morales, Fernández & Baca-González, 2017*), and similar results have recently been found when using MaxEnt for land-cover classification (*Mack & Waske, 2017*). However, our results show that although in general the identified best parameters outperformed the default parameters, for some land-covers and thresholds, default parameters can produce similar or even better classification results than the identified best parameters. Technically this sounds illogical, because a manually parameterized model can theoretically use the same parameters of the default model, and therefore if best model parameters are found, classification results should be at least equally good as the one obtained by the default model. Nevertheless, finding the best parameters combination is still a challenging task (*Muscarella et al., 2014*). For example, to find the optimal parameters

combination for our study, we run thousands of models and selected each parameter based on the median value of resulting *kappas*, assuming that general central tendencies could be indicative of parameters performances. However, this approach ruled out potential parameters combinations that produced higher *kappas* (See supplementary material), which may help explaining why in our study for some comparisons the default parameter had better classification accuracies than the best model parameter.

An interesting finding from our study was the fact that threshold selection do not only affect the accuracy of classification results (*Mack & Waske, 2017*), but that a same threshold could differentially impact the accuracy of different models (Fig. 6A). These findings suggest that using the model performance information (e.g., Area Under the Curve) for evaluating the accuracy of a land-cover classification outcomes is not advisable.

## Best parametrization practices

In the light of our results and other articles regarding the use of MaxEnt, we would like to highlight the importance of parameterizing MaxEnt for each model individually and avoid the temptation to of using default parameter or previous studies parametrizations (*Merow, Smith & Silander, 2013*; *Mack & Waske, 2017*; *Morales, Fernández & Baca-González, 2017*). As a rule of thumb, a comparison of different models with different parametrization settings must be performed to determine the best model among all the evaluated parameters combinations. Ideally, selection of parameters combination to be tested should not be random, but based on the specific objectives and land-cover under analysis. For example, for contrasting and homogenous land-cover such as paved areas, using combinations that generate simpler models could be an efficient approach. Whereas for less contrasting and heterogeneous land-covers, such as irrigated grass, it would be recommended to increase the number of tested combinations to include more complex models. There are several techniques available for selecting MaxEnt best models, including jackknife procedures based on the corrected Akaike information criterion (AICc) (*Shcheglovitova & Anderson, 2013*; *Morales, Fernández & Baca-González, 2017*), correlation analysis (*Syfert, Smith & Coomes, 2013*), fuzzy *kappa* statistics (*Mestre et al., 2015*) and software including specific algorithms for this objective (*Warren, Glor & Turelli, 2010*; *Muscarella et al., 2014*). Nevertheless, once the parametrization of the model is done, thresholds to generate binary maps still need to be chosen. As our results show, threshold selection could be one of the most important factors influencing the final classification accuracy. In this regard, we recommend not using predefined thresholds or thresholds used in previous studies, but trying a set of different thresholds and select the one that gives the best outcome. A viable option could be using a set of different thresholds based on cumulative percentiles and select the generated model with the largest classification accuracy (*Lin et al., 2014*). Based in our results, percentiles between 1 and 10 seems to be a good testing ground, as best classification outcomes tend to be produced by thresholds within this percentile range. Nevertheless, further research is needed to better understand how variation in threshold values affects the accuracy of land-cover classification produced through MaxEnt.

## CONCLUSION

Our results showed that MaxEnt can be a useful tool for single-class land-cover classification. However, the classification accuracy heavily depends on land-cover characteristics and parameters used for modelling. Based on these results, we suggest avoiding the temptation of using MaxEnt default parameterization features with heterogeneous layers, as manual parameterization could generate better land-classification accuracy outcomes. In addition, selection of optimal binary thresholds is key for increasing classification accuracy when using binary maps. Nevertheless, it is still essential to use a classification accuracy metric for testing modelling results. In light of these results, we suggest testing different parametrization settings and thresholds until satisfactory *kappa* or other accuracy metrics are achieved. Finally, is also important to bear in mind that the reliability of the selected model will be also influenced by the accuracy metric used for classification testing. In this regard, for specific classification tasks, it would be recommendable to use multiple accuracy metrics to evaluate the classification accuracy of the resulting models.

### Funding

The authors received no funding for this work.

### Competing Interests

The authors declare there are no competing interests.

### Author Contributions

- Ignacio C. Fernández conceived and designed the experiments, performed the experiments, analyzed the data, prepared figures and/or tables, authored or reviewed drafts of the paper, approved the final draft.
- Narkis S. Morales performed the experiments, analyzed the data, authored or reviewed drafts of the paper, approved the final draft.

### Data Availability

Raw data are available as Supplemental Files.

### Supplemental Information

Supplemental information for this article can be found online at http://dx.doi.org/10.7717/peerj.7016#supplemental-information.

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
