# Peer review of "One-class land-cover classification using MaxEnt: the effect of modelling parameterization on classification accuracy"

_PeerJ, doi:10.7717/peerj.7016_

## Round 0.1 · original submission · Minor Revisions

Two referees have provided positive reviews of your submission. The review comments indicate some issues that require attention but only relatively minor revision that should enhance the final paper. I have only one issue to flag in addition to the comments from the referees - the use of kappa coefficients can be problematic. Kappa is widely used but its magnitude can be greatly influenced by, for example, variations in prevalence. Some comments that the accuracy metric should be selected and interpreted with care would probably be sufficient to address the concern given the way the argument is made in the paper.

Reviewer 1 ·

Basic reporting

The authors present a systematic analysis about how modeling parameterization affect the classification accuracy of MaxEnt; if these parameter effects depend on the land-cover under analysis; and can we identify the best parameter setting for different land-cover classes and compare its performance with the default parameter setting. Overall, the study goals are well clear and logical. The article is well written, which is well structured and easy to follow. But the tables and figures still need to be improved.

Experimental design

The goal of this study is clear, and the experimental design is clear as well. But the results section need to be improved. Please see below for detailed comments.

Validity of the findings

Need to be improved. Detailed comments please see below.

Additional comments

General comments:
The authors present a systematic analysis about how modeling parameterization affect the classification accuracy of MaxEnt; if these parameter effects depend on the land-cover under analysis; and can we identify the best parameter setting for different land-cover classes and compare its performance with the default parameter setting. Overall, the study goals are well clear and logical, and the article is well written. However, the result and conclusion of the study are not very clear. Some major comments are showed as follows:
1. As mentioned in Line 15-17, the classification accuracy is highly affected by the land-cover type being classified. However, the land-cover classes in this study seems to be too few. In addition, the class level of land-cover is too simple/coarse (e.g., the build-up includes many types of land-covers), which is not clear to thoroughly analysis the effects of modeling parameterization, especially in spatial heterogeneous areas. For example, the FC (feature classes) parameter is important on the final accuracy (Line214) and heavily dependent on the land-cover. Of which, L and Q features are both simple features, which are beneficial to spatially homogenous covers (e.g., built-up), but weaken for more heterogeneous covers (e.g., irrigated). However, for the weakness, we do not know the reason due to heterogeneous covers. To make the parameter effect on different land-cover classifications, it would be better to analysis the parameter effects with more land-cover types.
2. Line 23, the SS (sampling size) has a positive effect on classification result. However, you did not give us the effect will exist to what extent. According to Fig.3, with the increase of SS, the K accuracy will increase. Therefore, we wondering whether you should to increase the sampling size from 100 to a larger value until the K value becomes stable. The comments are also suited for TP and RM parameters (Fig.2 and Fig.4).
3. Except the setting of SS, TP, and RM parameters, it is not clear if there are some specific FC and BT parameters that can provide higher quality classifications based on the results (Line 301, Line 336-339). In other words, users will need to explore the optimal value of the most two important parameter by continuous trying. Therefore, how can results of this study give some guidance for setting the two parameters?
4. Line 266-270, only around half comparisons (23/44) show that the best parameters outperformed the default parameters. It is really sounds illogical. The best parameter should possess best accuracy, at least equally well. Although you have given some discussions, which seems reasonable for the insignificant difference in the comparison result. However, for significant difference between the best parameter and default setting (e.g., MTP in fig.6a), it need more discussions.
Minor comments:
1. Line 175, what is the class distribution of the randomly selected 1000 points?
2. Line 50, “another issue” should be replaced by another word, such as however. Because from Line41 to Line 50, the content is about the strength of maximum likelihood algorithm.
3. The Abstract did not show the clear result of the three goals of the study. Some specific parameter setting guidance should be given.
4. Line 18, the sentence “but for heterogeneous land-cover …” is not result.
5. Line 309, “heterogenous” should be “heterogeneous”, please check thoroughly.
6. Please give full name of the abbreviations AUC, IF and NM (Line 365, Line 151, and Line185)
7. Line 395-397. You suggest using manual parameterization could generate better land-classification accuracy outcomes. However, it is not consistent for irrigated grass (Line 275-277).
8. Line 101, “depends” should be “depend”

·

Basic reporting

The article is well written and well structured, but a few sentences should be revised language-wise. Most of the relevant literature is cited, and the context is provided for. The submission is ‘self-contained’.

It is not clear to me whether or not the raw data is being shared / already available somewhere.

Experimental design

The article is original primary research that falls within the aims and scope of the journal. It clearly fills a knowledge gap, which is well identified. The methods are sufficiently described to be replicable.

Validity of the findings

The findings are valid and meaningful, and the benefit to the literature is clear; data seems to be robust and the conclusions are well stated.

Additional comments

The article tests the effects of varying Maxent modelling parameters as well as parameters related to the modelling process. The large number of different models that were built do allow to draw conclusions on the effects of those parameters, specifically for the application of using Maxent for land cover classification. As modelling algorithms are frequently applied without in-depth analysis of the effects of the modelling parameters used, the contribution is very valuable to anyone applying Maxent for land cover classification (or for species distribution modelling in general).

Here are some comments and suggestions for improvement:

Introduction
Line 70/71: “it has shown good performance” – Here it would be good to add some literature sources that have confirmed the good performance of Maxent, and maybe to add a sentence on why you chose to work with maxent because many other classifiers have also shown good performance and are widely used for land cover classification, e.g. SVM or regression trees.
Line 77/78: “the only published study on Maxent parametrization for land cover classification of which we are aware” – in a recent publication (Skowronek et al. 2018) we used Maxent to map an invasive species and tested the effect of varying feature class and regularization multiplier. The results were somewhat similar to your findings (see Skowronek S, Van De Kerchove R, Rombouts B et al (2018) Transferability of species distribution models for the detection of an invasive alien bryophyte using imaging spectroscopy data. Int J Appl Earth Obs Geoinf 68:61–72. https://doi.org/10.1016/j.jag.2018.02.001).

Methods
Line 131: “as MaxEnd models can be sensitive to over-parametrization” – please provide a reference for this statement; a statement in Elith et al. 2011 suggests the opposite, mentioning that Maxent is rather robust and possibly able to deal with correlated input data. (“In particular, it is more stable in the face of correlated variables than stepwise regression, so there is less need to remove correlated variables” (see Elith, J., Phillips, S.J., Hastie, T., Dudík, M., Chee, Y.E., Yates, C.J., 2011. A statistical explanation of MaxEnt for ecologists. Divers. Distrib. 17, 43–57. http://dx.doi.org/ 10.1111/j.1472-4642.2010.00725.x.).
Line 158: You varied the sample size from 40 to 100 samples – I assume this is the number of presence points? What about the background points? How many background points did you use, how did you generate them, or did you use the respective presence points of the other classes as background? As the choice of the background plays an essential role this should definitely be explained.
Line 168: “All MaxEnt models were run using the random seed feature” – please provide a short explanation what the random seed feature is or a reference / where it is availabe.
Method section: Do you say somewhere what program/ software packages you used for the modelling? If not already included please add this information. Might also be helpful to include your full script as supplement.
Line 185: “IF and NM” – I assume those are the initials of the authors, maybe substitute the abbreviations by “the authors of this study” or the full names; the abbreviations are confusing.
Whole manuscript: You are using the abbreviations for fc, rm, ss, tp etc. throughout the text. However, these abbreviations are not really standard, or well known abbreviations, especially not to readers who have not used Maxent before. I suggest using the full expressions (feature class, regularization multiplier etc.) as much as possible, and only using abbreviations where really necessary, e.g. in graphics where you don’t have enough space, or in equations. Most people are not able to remember several previously unknown abbreviations along the whole article; it makes some parts of the text unnecessarily difficult to.

Results
Line 219/220: “was the largest differed” – please revise the sentence language wise
Line 223/224: “T, Q and TQ” – shouldn’t it be L, Q and LQ?
Line 261-266: sentence is difficult to read, please revise sentence structure.
Line 361-365: please revise language wise

Discussion:
Line 329-339: From your data, is there any conclusion you can draw on what type of threshold worked better for which land cover class? Or can you suggest what research would be necessary to approach this topic?
Also, there are a few more publications that have varied fc and rm in studies aimed at modelling species distribution, some of them could be included (see discussion in Skowronek et al. 2018).

Supplement
The supplements contain relevant tables showing an overview of the modelling results and parameters. However, the excel worksheet “Graphs” is only telling me that “this chart is not available in your version of excel”; I would suggest exporting the graphs as PDF instead of using Excel.

---

## Round 0.2 · accepted · Accept

The revisions made appear to have enhanced the article. I personally still have concerns on the use of kappa but would not wish to let my views on kappa unduly impact on the review process. I hope that your article is well-received.